# Knowledge, Vaccination Status, and Reasons for Avoiding Vaccinations against Hepatitis B in Developing Countries: A Systematic Review

**DOI:** 10.3390/vaccines9060625

**Published:** 2021-06-09

**Authors:** Putri Bungsu Machmud, Saskia Glasauer, Cornelia Gottschick, Rafael Mikolajczyk

**Affiliations:** 1Institute of Medical Epidemiology, Biometrics and Informatics (IMEBI), Interdisciplinary Center for Health Sciences, Medical School of the Martin-Luther-University Halle-Wittenberg, Magdeburger Straße 8, 06112 Halle (Saale), Germany; putri.machmud@uk-halle.de (P.B.M.); saskia.glasauer@uk-halle.de (S.G.); cornelia.gottschick@uk-halle.de (C.G.); 2Department of Epidemiology, Faculty of Public Health, Universitas Indonesia, Jl. Prof. Dr. Bahder Djohan, Depok 16424, Indonesia

**Keywords:** developing countries, hepatitis B, knowledge, vaccination status, risk population

## Abstract

(1) Background: The coverage of hepatitis B vaccination remains low in developing countries to date. This systematic review thus analyzes the determinants of people’s knowledge and vaccination status as well as the reasons why people in developing countries chose not to receive the hepatitis B vaccination. (2) Methods: We searched four databases to identify all studies from developing countries published within the past 10 years. Both low-risk and high-risk populations aged older than 15 years old were eligible for the study. The quality of studies was assessed by the Newcastle–Ottawa Scale assessment. (3) Results: This study identified 2443 articles, 89 of which were included in the analysis. Monthly income, occupational status, and profession as a health-care worker were the strongest predictive factors for both knowledge of hepatitis B and vaccination status. In addition, strong predictor variables of hepatitis B knowledge were knowing an infected person and level of education, while health insurance, management’s protection at workplace, infection training, and experience of hepatitis B exposure were strong influencing factors for vaccine uptake. (4) Conclusions: Exposure to information, support from institutions, and financial support related to vaccination cost have a positive impact on the knowledge about hepatitis B infection and vaccination coverage.

## 1. Introduction

In 2020, the World Health Organization (WHO) reported that 325 million people worldwide were living with chronic hepatitis B infection and approximately 900,000 people died due to hepatitis B [1]. The majority of cases, 68%, were recorded in the African and Western Pacific regions [2].

Vaccination is considered the most effective way of hepatitis B prevention. Nevertheless, the coverage of hepatitis B vaccination remains low in developing countries [3,4,5,6]. The countries’ inability to face the hepatitis B burden due to political and financial problems may, thereby, pose a substantial obstacle to prevention [7]. For instance, previous studies found that only 33% of health-care workers (HCW) in Tanzania and 23% of the general population in Korea were fully vaccinated against hepatitis B [3,8]. Potential explanations for these findings include lack of knowledge and awareness of hepatitis B. A systematic review among immigrants and refugees residing in the US, Canada, and Australia by Owiti et al. showed that vaccine acceptance and people’s attitude towards hepatitis B was associated with their knowledge about the disease [9]. Furthermore, Abiodun et al. reported that more than 70% of hospital cleaners in Nigeria failed to recognize all ways of hepatitis B transmission and prevention and named the lack of awareness of hepatitis B as a reason not to be vaccinated [4].

Previous studies from developing countries indicate a variety of factors predicting the level of knowledge and realization of hepatitis B vaccination in adults, but with mixed results. While some conclude that sociodemographic factors, such as age and marital status, are associated with the level of knowledge and vaccination status of hepatitis B [6,10,11,12,13,14], other studies could not find such results [15,16,17,18,19,20]. As a result, this systematic review aims to summarize the available evidence in order to identify predictors of the level of knowledge and vaccination status of hepatitis B and reasons why people chose not to be vaccinated against hepatitis B in developing countries.

## 2. Materials and Methods

### 2.1. Protocol Registration and Reporting Structure

A protocol for this review was registered in the international prospective register of systematic reviews (PROSPERO) with the registration number CDR42020179001 [21]. This manuscript was written using the Preferred Reporting Items for Systematic reviews and Meta-Analyses statement [22,23] (Appendix A).

### 2.2. Eligibility Criteria

Inclusion criteria for this review were determined according to the population, intervention, comparator, outcome, and setting (PICOS) format. The population consists of all adults, including both the population at low and high risk of contracting hepatitis B. People at high risk were defined as people who live and/or work or study closely with hepatitis B patients, including health-care workers (HCW), students in a medical or health-related fields (medicine, dentistry, nursing, and midwifery), pregnant women, barbers, municipal workers, and sex partners and household members of people with hepatitis B. The low-risk population was defined as people from the general population, being >15 years old and not living or working/studying closely with hepatitis B patients. There was no intervention or comparator in this study. Outcomes were the level of knowledge and vaccination status of hepatitis B and the setting was developing countries. We followed the list of developing countries as published in the official report from the United Nations in 2020 [24]. All study designs published within the past 10 years were included. Studies for which the full text was not available in English, abstracts from conferences, and systematic reviews were excluded from this review.

### 2.3. Databases and Search Strategy

We searched four databases: MEDLINE, Embase, Web of Science and CINAHL [25]. The search began by listing the keywords through MeSH terms: “Health personnel”*, “Healthcare worker”*, “Healthcare provider”*, Patient*, Student*, Person*, Adult*, Knowledge*, Practice*, Vaccine*, Uptake* Vaccination*, Immunization*, Immunisation*, Campaign* and Hepatitis B*. Furthermore, we combined the list of keywords using OR and AND in the advanced search (Appendix A). Duplicate articles were checked using the Endnote8 system. Articles were first screened based on the title, followed by abstract und full paper screening (Figure 1).

### 2.4. Data Extraction and Management

Selected articles were extracted using a standard table format and entered into Microsoft Excel. The extracted data included information on country and region of study, author and publication year, population under investigation (pregnant women, general population, students and HCW), study design, sample size, percentage of good knowledge, and proportion of vaccine uptake per dose.

### 2.5. Risk of Bias Assessment

Two independent reviewers (P.B.M. and S.G.) performed quality assessment. Rating and scoring were conducted using the Newcastle–Ottawa Scale (NOS) checklist for quality assessment [26]. Articles were divided into three categories of quality: unsatisfactory, satisfactory, and good study. Studies were considered unsatisfactory when they had a score of less than five for cross-sectional studies and less than four for cohort/case-control studies. Cross-sectional studies scoring 5 to 7 or cohort/case-control studies scoring 4 to 6 were considered satisfactory. Studies were categorized as good when they reached a score of more than 7 for cross-sectional and more than 6 for cohort/case-control studies (Appendix A).

### 2.6. Data Analysis and Synthesis

Data were analyzed separately for the outcomes knowledge of hepatitis B and vaccination status of hepatitis B vaccination. Independent variables were age, gender, education, residency, marital status, monthly income, ethnicity, occupational status, and health insurance for sociodemographics; HCW profession, part-time job, work department, work experience, work regimen, and level of satisfaction with the profession; facility level, management protection at workplace for workers; year of study, and type of facility (university or faculty) for students. In addition to the aforementioned variables, we considered information on exposure to hepatitis B (ever joined in training on infection diseases, and ever heard about hepatitis B before) as well as exposure to hepatitis B, previous hepatitis B screening, and alcohol and tobacco consumption. Other than that, vaccination status of hepatitis B, hepatitis B knowledge and information on the reasons why people chose not to immunize are provided in this review.

## 3. Results

Of 2443 articles, 445 were removed due to duplication and 1812 were eliminated, as they did not meet the inclusion criteria such as children’s immunization, setting in developed countries, and systematic review paper. Furthermore, 98 articles were excluded for the following reasons: 18 were conference abstracts, 10 articles were non-English, and 69 articles had irrelevant outcomes. As a result, 89 articles met the eligibility criteria, resulting in an overall sample size of 73,988 participants (Figure 1).

### 3.1. Characteristics of Included Studies

More than 90% (*N* = 83, 93.3%) of the studies were conducted in Asia and Africa, and 48.3% (*N* = 43) were hospital/health facility-based studies. Furthermore, 80.9% (*N* = 72) of the studies included high-risk population and 46.1% (*N* = 41) of studies were about HCW. Most studies (*N* = 87, 97.8%) were cross-sectional studies, with the remaining (*N* = 2, 2.2%) being case-control and retrospective cohort studies. Fifty-eight studies (65.2%) included knowledge of hepatitis B as an outcome and 69 studies (77.5%) assessed the uptake of hepatitis B vaccination (Table 1).

Forty-seven of the 58 studies (81.0%) analyzing knowledge of hepatitis B included a population at high risk. Among these, 24 studies (51.1%) were on HCW, 18 studies on students in a medical or health-related field (38.3%), 3 studies (6.4%) on pregnant women, and two studies (4.3%) on others. Of the 69 studies that addressed hepatitis B vaccination status, 59 (85.5%) were based on the high-risk population. Again, most were on HCW (N = 35, 59.3%), followed by students in the medical field (*N* = 20, 33.9%).

### 3.2. Study Quality

Regarding the methodological quality, most of the studies that assessed knowledge of hepatitis B (*N* = 45, 77.6%) were rated as unsatisfactory, and 12 studies (20.7%) were considered satisfactory. Only one study (1.7%) was of good quality. Similarly, 52 studies (75.3%) analyzing the uptake of hepatitis B vaccination were rated to have unsatisfactory quality; 14 studies (20.3%) were considered satisfactory and, again, only three studies (4.3%) were of good quality.

Only four cross-sectional studies stated clearly how risk factors and exposures were ascertained [16,27,28,29], and that vaccination status was recorded based on personal recall and confirmed by a vaccine registry of hospital or clinic registry. In addition, only 23.0% (*N* = 20) of studies provided information on the proportion of the target sample recruited or a basic summary of non-respondent characteristics. We also found that only 37 studies (42.0%) used statistical tests for analysis, provided a clear description of such, and presented the strength of the association including the confidence interval.

### 3.3. Hepatitis B Knowledge

Thirty-three studies (56.9%) reported the level of knowledge using percentages of correct answers while ten studies (17.2%) provided means. Five studies (8.6%) reported both percentages and means. Generally, there is a great diversity among the studies in the definition of cutoff points for good knowledge. For example, a survey from Malaysia defined good knowledge about hepatitis B based on the 75% cutoff point (17 or more out of 22 questions correctly answered) [10], while Chung et al. categorized answers into adequate and inadequate knowledge based on a 85% cutoff point [15]. Ahmad et al. used the median as the cutoff point for assessing the level of knowledge [5].

The reported proportion of people with good or adequate knowledge ranged from 1.1% to 83.8% in the high-risk population and from 17.0% to 50.3% in the low-risk population (Table 2). Students were found to have the highest proportion of good or adequate knowledge of hepatitis B among the populations at risk. The median proportion of having good or adequate knowledge was 63.5% (IQR 47.8–77.5%) and 37% (25.5–43.5%) among high- and low-risk populations, respectively (Figure 2).

### 3.4. Hepatitis B Vaccination

The median proportions of getting at least one dose and getting complete doses of hepatitis B vaccination among the high-risk population were 50% (IQR 34.5–73%) and 39% (IQR 21.3–58%), respectively. The median proportion of getting at least one dose and complete doses of hepatitis B vaccination among the low-risk population were 37% (35–74.5%) and 27% (19.3–34.8%), respectively (Figure 3). Therefore, populations at high risk tend to have a higher proportion of complete vaccination (median percentage 39.1% vs. 27%) than the low-risk population. In addition, HCW were found to have the highest proportions of both receiving at least one dose and receiving the complete vaccination among the populations at high risk (Figure 3).

### 3.5. Factors Associated with Knowledge and Vaccination Status

Overall, variables which predicted the knowledge and vaccination status of hepatitis B could be summarized in eight categories: sociodemographic, work related to hepatitis B, student related to hepatitis B, information exposure, exposure experience, vaccination status, knowledge of hepatitis B, and lifestyle in the high- and low-risk populations (Table 3 and Table 4).

Nine of 58 (15.5%) studies addressing knowledge and 22 of 69 (31.8%) studies addressing vaccination status assessed sociodemographic factors as predictor variables. Among these, monthly income and level of education were strong predictors of both hepatitis B knowledge and vaccination status in both the high- and low-risk population.

In the high-risk population, four (6.9%) and 15 (22.1%) studies discussed the association between work-related factors and hepatitis B knowledge and vaccination status, respectively. However, only profession as a HCW influenced hepatitis B knowledge and vaccination status among participants. In addition, management’s protection at workplace was a predictor for vaccine uptake. Apart from that, of the four studies assessing the association between being a student and hepatitis B knowledge and vaccination status, only one study found that year of study and type of university or school were predictor variables for hepatitis B knowledge [6,17,92,99].

Two (3.4%) and five (7.4%) studies analyzed the association between exposure to information and hepatitis B knowledge and vaccine uptake, respectively, and found that ‘ever heard about hepatitis B’ had a positive association with better knowledge in both the high- and low-risk populations [92]. Furthermore, among HCW, four of five studies showed that experience in infection training on hepatitis B was a strong predictor variable for vaccination status [29,35,67,90].

Knowing an infected person, screening for hepatitis B, family history, and exposure to hepatitis B were variables included in twelve studies related to knowledge and vaccination status of hepatitis B. Of those, exposure and previous hepatitis B screening influenced vaccine uptake while knowing an infected person was a strong predictor variable for hepatitis B knowledge in the low-risk population. In addition, in the high-risk population, vaccination status influenced hepatitis B knowledge [11,99], while knowledge of hepatitis B [90] and lifestyle (alcohol and tobacco used) [52,57,70] were predictors for vaccine uptake, albeit on a lower level.

### 3.6. Reasons for Not Being Immunized

In this systematic review, 32 studies (36%) assessed people’s reasons for not being immunized, of which most (*N* = 29, 90.6%) were conducted in the high-risk population (left side of the red line). Here, among HCW (*N* = 17, 53.1%), the most common reason for not being vaccinated was vaccine costs (*N* = 12, 70.6%) [4,16,31,32,35,44,56,66,67,72,77,90], followed by lack of time (*N* = 10, 58.8%) [11,16,35,44,56,63,66,67,77,90] and lack of motivation (*N* = 9, 52.9%) [16,35,39,44,56,66,77,80,90], including ‘never felt the need for vaccination’ or ‘having no fear of catching hepatitis B infection’. Slightly different results were found among students of medicine or a health-related field (*N* = 10, 31.6%) [6,33,50,58,60,65,73,74,83,99]. Here, lack of motivation was the major reason against vaccination (*N* = 8, 80%) [6,33,50,58,60,65,73,74,83,99], followed by fear of injection or side effects (*N* = 5, 50%) [6,58,60,65,99], and lack of information (*N* = 5, 50%) [6,33,73,74,99]. Three studies (9.4%) addressed the low-risk population, which is general population (right side of the red line) and found three major reasons: lack of motivation (*N* = 2, 66.7%) [19,20], lack of time (*N* = 3, 100%) [18,19,20], and lack of information (*N* = 2, 66.7%) [18,19] (Figure 4).

## 4. Discussion

We conducted a comprehensive systematic review of hepatitis B knowledge and vaccination status and predicting factors in developing countries that included articles published between 2010 and 2019. Overall, 2443 records were identified and 89 articles were ultimately included. Of these, 58 and 69 studies provided data on knowledge and vaccination status of hepatitis B, respectively.

### 4.1. Main Findings

We found that important determinants for the level of knowledge and vaccination status varied considerably across studies. However, the strongest predictive factors for hepatitis B knowledge and vaccination status were monthly income, level of education, and profession as HCW in the high- and low-risk populations. Being ever screened for hepatitis B was a strong influence for hepatitis B knowledge, while health insurance, management’s protection at workplace, experience in infection training on hepatitis B, and experience of hepatitis B exposure were strong predictors for vaccine uptake.

We also revealed that there are different predictors of the level of knowledge and vaccination status between high-risk and low-risk populations. This is most likely due to a limited number of studies regarding the low-risk population (10%). Among those, only two and three studies assessed the predictor variables of hepatitis B knowledge and vaccination status, respectively. Additionally, some variables were only assessed in a particular population, e.g., variables related to work were only collected among HCW.

Other than that, this study found that lack of motivation, lack of information, and lack of money were three major reasons for people to avoiding hepatitis B vaccination in developing countries.

### 4.2. Factors Associated with Knowledge and Vaccination Status

Whereas sociodemographic variables such as age, sex, ethnicity, residency, and marital status were still inconclusive predictor variables, monthly income and level of education were found to be strong predictors for hepatitis B knowledge and vaccination status.

A number of articles demonstrated that participants with higher income had better knowledge and vaccination uptake [10,13,15,18,19]. This is reasonable, because income is a driving force behind the health disparities and is directly associated with health literacy. Hence, according to Tang et al., people with lower income are less likely to seek health information or to use health-care professionals as a first source of health information and have greater difficulty understanding information compared to people with higher income [104].

Another strong predictor for hepatitis B knowledge and vaccination status was education. Our review revealed that the higher the level of education, the more likely a person was to have good knowledge and to receive hepatitis B vaccination. This is most likely due to the fact that education affects health through an individual’s improved ability to acquire and process health-related knowledge, and improved health behavior [105].

Our findings also indicate that factors related to work, such as profession as HCW, had strong evidence as predictor variables for hepatitis B knowledge and vaccine uptake among the high-risk population. People working in high-risk conditions of hepatitis B transmission were more likely to have good knowledge of the disease and tend to protect themselves from infection through vaccination [15,17]. This is reasonable because HCW have wider access to information which has a substantial impact on both knowledge and vaccination status. Ochu et al., for example, revealed that the higher the perceived risk of contracting hepatitis B, the higher the awareness of the need for vaccination [102].

Furthermore, in the high-risk population, a workplace with good occupational protection measures in place most likely also had higher hepatitis B vaccination coverage among the employees. These protection measures could be in the form of mandatory use of personal protective equipment [14,80], provision of free hepatitis B vaccination for employees [90], and regular safety training for employees, including demonstration of the benefits of hepatitis B vaccination [52].

People with previous experience related to hepatitis B, such as people with family members or friends infected by hepatitis B, or people with a positive hepatitis B screening result, tended to have a better knowledge of the disease and received hepatitis B vaccination for prevention purposes [11,28,70,80]. Therefore, the greater the experience with or exposure to hepatitis B, the better knowledge people had and the greater their willingness to receive vaccination.

Information exposure has a direct association with good knowledge, which in turn, also has an impact on vaccination status in both high-risk and low-risk populations. Eni et al. found that persons who had ever heard about hepatitis B before were more likely to have been vaccinated and have a higher score of knowledge [92]. This finding is supported by Mungandi et al., who found that HCW who were ever trained in infection control were twice as likely to be vaccinated against hepatitis B as those who were not trained before [29].

A study among dentists in Monte Carlo reported that lifestyle factors, such as alcohol consumption and tobacco use, had a negative association with vaccine uptake [57]. The study estimated that non-smokers and people not consuming alcohol were 2.5 and 3.0 times more likely to receive the hepatitis B vaccine, respectively [57]. Correspondingly, a variety of other studies found a lower prevalence of vaccination among people consuming alcohol [52]. This association might be explained by increased health awareness, as people with a healthy lifestyle tend to protect themselves from any potential disease, including hepatitis B.

### 4.3. Reasons for Not Being Vaccinated

Generally, there were three major reasons for people opposing vaccination: lack of motivation, lack of information, and lack of money. However, some of these reasons were interrelated, as the lack of information and awareness of the vaccination might influence someone’s belief regarding its effectiveness. Poor information regarding hepatitis B infection and vaccination reduces people’s motivation to vaccinate, as most participants claimed they never felt the need to vaccinate against hepatitis B infection. Although none of the selected studies reported fear of occult hepatitis B infection after vaccination as a reason not to vaccinate among the adult population, Aghakhani et al. found that hepatitis B vaccine escape mutants had caused infections in vaccinated individuals since 1990s. This issue might be considered as another factor influencing vaccine hesitancy [106,107,108]. Hence, there is a pressing need for information about hepatitis B infection, benefits of hepatitis B vaccination, and the emergence of vaccine escape mutations through, e.g., participation in infection training on hepatitis B regularly, and increasing risk perception and awareness of hepatitis B vaccination among the adult population, especially for high-risk populations such as HCW.

Furthermore, according to Park et al., the lack of awareness is the main barrier to vaccination in the population [18]. Apart from that, HCW named the costs of the hepatitis B vaccination as the most common reason against it. Given that HCW are a group at high risk of contracting hepatitis B and can also take the role of a vector in the transmission of disease to their patients, health-care systems should advocate health policies for free hepatitis B vaccination for HCW. For example, a system could be implemented in the workplace that provides management protection for staff such as free vaccinations.

### 4.4. Strength and Limitation of the Review

By focusing on developing countries, this study attempted to identify specific patterns from more than 70% of the world’s population. This is considered essential in providing a considerable amount of information about relevant variables of hepatitis B knowledge and vaccination status in a variety of countries with different cultures and financial abilities to run health programs. Apart from that, there are some limitations to this review. First, most studies (77.6%) were considered unsatisfactory, because they did not assess outcomes in a multivariable analysis, identifying important predictor variables for knowledge about hepatitis B and vaccination status. Second, approximately 82% of studies were based on high-risk populations such as HCW. Third, our study did not look at vaccine escape mutations regarding any population criteria due to limited scientific evidence. Therefore, we report the results stratified by population allowing interpretation for both the high- and low-risk populations. Last, the search strategy was restricted to papers that were peer reviewed and written in English and, thus, 10 included articles written in the countries’ mother tongues, such as Mandarin, French, and Turkish were missed.

## 5. Conclusions

Our results suggest that various factors are associated with knowledge and vaccination status relating to hepatitis B. Some of the variables showed a strong and consistent relationship, while findings regarding others were inconclusive. In addition, there were different predictor variables for hepatitis B knowledge and vaccination status in the high-risk and low-risk populations. Exposure to information has an impact on increasing knowledge and awareness of hepatitis B infection and vaccination. In addition, institutional support, i.e., from the workplace, is needed through management protection for employees and especially for those at high risk. Finally, financial support related to vaccination is an important factor in increasing vaccine coverage. Therefore, stakeholders could improve further hepatitis B vaccination programs and research by providing funds for routine monitoring and evaluation of vaccination coverage as well as research funding. For this purpose, our review can act as guideline on important factors for prioritization in vaccination programs. In addition, further studies of good quality are necessary to improve the ascertainment of risk factors, using vaccine records or vaccine registries instead of personal recall only.

## Figures and Tables

**Figure 1 vaccines-09-00625-f001:**
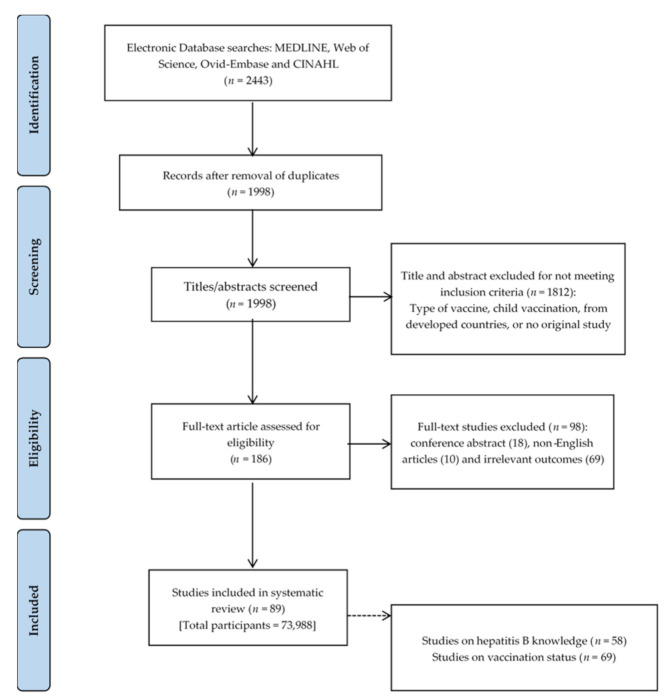
Flowchart of study search (adopted PRISMA: 2009).

**Figure 2 vaccines-09-00625-f002:**
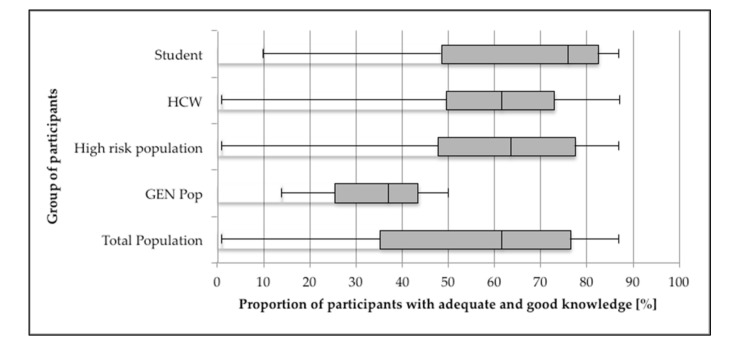
Proportion of participants with good and adequate knowledge combined.

**Figure 3 vaccines-09-00625-f003:**
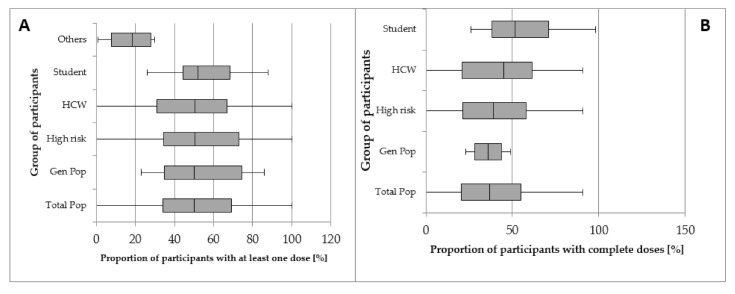
Proportion of participants with (**A**) at least one dose and (**B**) complete doses of hepatitis B vaccination.

**Figure 4 vaccines-09-00625-f004:**
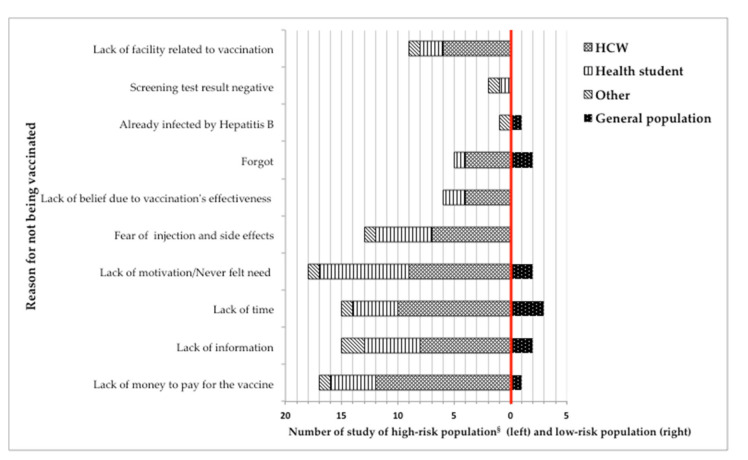
Number of studies addressing reasons for not being vaccinated by high-risk and low-risk populations. ^§^ High-risk population consisted of HCW, students, and others.

**Table 1 vaccines-09-00625-t001:** Summary of studies.

Variable	Categories	Number of Studies (%)
Year of publication	2010	7 (8%)
	2011	7 (8%)
	2012	5 (6%)
	2013	6 (7%)
	2014	10 (11%)
	2015	8 (9%)
	2016	9 (10%)
	2017	10 (11%)
	2018	15 (17%)
	2019	12 (13%)
Region	Central Africa	5 (6%)
	East Africa	8 (9%)
	East Asia	16 (18%)
	North Africa	2 (2%)
	South America	6 (7%)
	South Asia	24 (27%)
	Southern Africa	2 (2%)
	West Africa	19 (21%)
	Western Asia	7 (8%)
Year of study	2004–2008	14 (16%)
	2009–2013	22 (25%)
	2014–2018	38 (43%)
	N/A	15 (17%)
Design study	Cross-sectional	87 (98%)
	Case control	1 (1%)
	Cohort retrospective	1 (1%)
Study sites	Hospital/health facility-based	43 (48%)
	Institution-based	34 (38%)
	Community-based	12 (13%)
Population	High-risk population	72 (81%)
	Low-risk population	16 (18%)
	High- and low-risk population	1 (1%)
Participant *	Health-care worker	41 (46%)
	Student	29 (33%)
	Pregnant women	5 (6%))
	General population >15 years old	9 (10%)
	Others **	5 (6%)
Approached	Interviewed	18 (20%)
	Self-administered	55 (62%)
	Interviewed and self-administered	1 (1%)
	N/A	15 (17%)
Quality grade (knowledge) (*n* = 58)	Unsatisfactory studies (US)	46 (79%)
	Satisfactory studies (SS)	11(19%)
	Good study (GS)	11(19%)
Quality grade (vaccination status) (*n* = 69)	Unsatisfactory studies (US)	52 (75%)
	Satisfactory studies (SS)	14 (20%)
	Good study (GS)	3 (4%)
Outcome *	Knowledge	58 (65%)
	Practice of vaccination	69 (78%)

* Some articles have more than one participant/outcome; ** Teacher, household contact, sexual partner, barber, municipal worker, migrant worker, and conference participant.

**Table 2 vaccines-09-00625-t002:** Overview of hepatitis B knowledge and vaccination status.

First Author	Year of Publication	Country	Participants	Sample Size (Participant Rates)	Good Knowledge		Vaccination Status		
					Total Score	NOS Score	At Least One Dose	Complete Dose	NOS Score
**High-risk population**									
Aaron [3]	2017	Tanzania	HCWs	334 (96%)	-	-	57%	34%	5
Abeje [30]	2015	Ethiopia	HCWs	354 (88%)	62%; 7.6 ± 1.27 ^a^	2	10%	5%	2
Abiodun [4]	2019	Nigeria	Cleaner worker in hospital (HCW)	89 (91%)	1.1%; 1.1 ± 1.8 ^a^	2	0%	0%	2
Abiola [31]	2016	Nigeria	HCWs	134 (94%)	57%; 72.5 ± 7.6 ^a^	4	-	49%	4
Abiola [32]	2013	Nigeria	HCWs	84 (96%)	70%	3	59%	13%	3
Adekanle [11]	2014	Nigeria	HCWs	382 (76%)	-	6	N/A	65%	6
Adenlewo [33]	2017	Nigeria	Medical and dental students	113 (94%)	-		83%	80%	1
Adjei [34]	2018	Ghana	Pregnant women	196 (89%)	6.1 ± 1.2 ^a^ (physician); 6.1 ± 1.9 ^a^ (midwife)	5	-	-	-
Adeyemi [17]	2013	Nigeria	Pregnant women	643 (100%)	24%	5	10%	-	5
Akibu [35]	2018	Ethiopia	HCWs	386 (97%)	-	-	-	26%	7
Al-Hazmi [36]	2019	Saudi Arabia	HCWs	41 (85%)	61%	2	58.5%	-	2
Alavian [37]	2011	Iran	Dental students	142 (89%)	-	1	-	-	-
Alese [38]	2016	Nigeria	HCWs	187 (NS)	-	-	16%	-	0
Ali [39]	2017	Pakistan	HCWs	381 (89%)	15.5 ± 3.69 ^a^	2	-	-	-
Alqahtani [40]	2014	Saudi Arabia	HCWs and health students	600 (100%)	87%	3	-	-	-
Aniaku [41]	2019	Ghana	Nursing training students	358 (NS)	30%	2	67%	50%	2
Aroke [42]	2018	Cameroon	Medical students	714 (94%)	83%	2	26%	17%	2
Asif [43]	2011	Pakistan	Medical students	375 (95%)	-	-	57%	50%	2
Assuncao [14]	2012	Brazil	HCWs	1770 (NS)	-	-	86%	75%	6
Attaullah [44]	2011	Pakistan	HCWs	824 (NS)	-	-	98%	73%	1
AydemiR [45]	2016	Turkey	HCWs	1359 (NS)	-	-	82%	-	1
Bedaso [46]	2018	Ethiopia	HCWs	241 (93%)	61%; 6.6 ± 0.9 ^a^	4	30%	22%	4
Bekele [47]	2014	Ethiopia	HCWs	98 (75%)	-	-	25%	18%	1
Celikel [48]	2014	Turkey	Pregnant women	198 (NS)	-	-	0.5%	-	2
Chan [13]	2011	Hong Kong	Pregnant women	1697 (85%)	Detailed per-question	4	-	-	-
Chao [49]	2010	China	Others	250 (NS)	13 (4–16) ^b^	3	-	-	-
Chingle [50]	2017	Nigeria	Medical students	1200 (NS)	-	-	48%	30%	4
Choudhary [51]	2017	India	Medical students	100 (NS)	82%	0	64%	-	0
da Costa [52]	2013	Brazil	HCWs	762 (96%)	-	-	-	53%	6
de Souza [53]	2014	Brazil	Medical students	675 (79%)	-	-	49%	-	0
Debes [54]	2016	Tanzania	HCWs	114 (NS)	-	1	35%	-	1
Demsis [55]	2018	Ethiopia	Medical students	408 (97%)	81%	6	-	-	-
Dev [56]	2018	India	HCWs	300 (66%)	-	2	34%	7%	2
Ferreira [57]	2012	Brazil	HCWs	292 (88%)	-	-	-	91.2%	5
Ghomraoui [58]	2016	Saudi Arabia	Medical students	444 (93%)	47%	4	88%	60%	4
Guerra [27]	2018	Brazil	Pregnant women	324 (NS)	-	-	26.8%	-	2
Hebo [59]	2019	Ethiopia	HCWs	230 (NS)	74%	4	-	-	-
Ibrahim [60]	2014	Syria	Medical students	128 (NS)	-	1	44%	-	1
Iqbal [61]	2019	India	Medical students	341 (NS)	-	-	55%	37%	0
Jaquet [12]	2017	Senegal	HCWs	127 (NS)	38 (34–44) ^b^	4	-	-	-
Joukar [62]	2018	Iran	HCW and others	3391 (58%)	-	4	-	-	-
Kesieme [63]	2011	Nigeria	HCWs	228 (NS)	-	1	27%	-	1
Khan [64]	2010	Pakistan	Medical students	1509 (NS)	10%	1	79%	55%	1
Khandelwa [65]	2018	India	Dental students	240 (NS)	-	2	45%	-	2
Ko [66]	2017	South Korea	HCWs	242 (44%)	-	-	100%	69%	4
Kouassi [67]	2017	Côte d’Ivoire	HCWs	291 (NS)	-	-	47%	-	4
Li [68]	2015	China	Dental intern students	313 (95%)	83.8%	2	-	-	-
Machiya [69]	2015	Botswana	HCWs	117 (59%)	17%; 7.9 ± 2.3 ^a^	4	50%	31%	5
Meriki [70]	2018	Cameroon	HCW and others	265 (NS)	-	-	30%	5%	5
Mirzaei [28]	2019	Iran	HCWs	299 (100%)	-	-	-	58.5%	7
Mungandi [29]	2017	Zambia	HCWs	331 (NS)	78%	-	19%	-	4
Mursy [71]	2016	Sudan	HCWs	110 (73%)	58%	2	73%	41%	2
Mustafa [72]	2015	Sudan	HCWs	372 (NS)	-	2	73%	-	2
Noubiap [73]	2013	Cameroon	Medical students	111 (NS)	83% (risk factor)	1	31%	18%	1
Noubiap [74]	2014	Cameroon	Surgical residents	49 (70%)	Detailed per-question	1	47%	25%	2
Ogoina [75]	2014	Nigeria	HCWs	290 (76%)	-	-	65%	-	3
Okwara [76]	2012	Nigeria	HCWs	169 (NS)	-	2	55%	31%	2
Omotowo [77]	2018	Nigeria	HCWs	3132 (91%)	-	3	51%	-	4
Oyewusi [78]	2015	Nigeria	HCWs	210 (88%)	65%	2	66%	-	-
Pathoumthong [6]	2014	Lao	Health students	961 (NS)	72%	5	31%	21%	6
Ray [79]	2017	India	Dental students	269 (NS)	76%	0	-	52%	0
Resende [80]	2010	Brazil	HCWs	1134 (87%)	-	-	74%	-	7
Rathi [81]	2018	India	Medical students	161 (81%)	-	2	-	-	-
Sandeep [82]	2010	India	HCWs	141 (82%)	7.3 ± 4.4 ^a^	3	-	-	-
Shahbaz [83]	2014	India	Medical and dental students	300 (NS)	-	1	40%	8%	1
Shukla [84,85]	2016	India	HCWs	89 (NS)	-	2	37%	-	2
Singh [86]	2011	India	Dental students	245 (NS)	-	2	39%	-	2
Tatsilong [16]	2016	Cameroon	HCWs	100 (61%)	47%	6	19%	-	5
Usmani [87]	2010	India	HCWs	215 (NS)	-	-	67%	51%	2
Vo [88]	2018	Viet Nam	Healthcare students	2017 (NS)	-	4	69%	-	4
Yamazhan [89]	2011	Turkey	Nursing students	1491 (89%)	-	5	85%	-	5
Yuan [90]	2019	China	HCWs	4168 (86%)	-	-	86%	60%	4
Zheng [91] †	2015	China	HCWs	1420 (NS)	-	-	40%	-	8
**Low-risk population**									
Ahmad [5]	2016	Malaysia	Students	662 (72%)	50.3%	3	-	14%	3
Chung [15]	2012	Hong Kong	General population >15 years old	1982 (90%)	14.0%; 13.5 ± 2.8 ^a^	5	63%	-	5
Eni [92]	2019	Nigeria	Students and general population >15 years old	758 (94%)	-	4	35%	-	3
Lee [93]	2010	South Korea	Students	711 (NS)	1.3 ± 1.7 ^a^	4	-	-	3
Moezzi [94]	2016	Iran	General population >15 years old	2956 (99%)	-	-	23%	21%	2
Mustufa [95]	2010	Pakistan	Teacher	200 (NS)	-	-	37%	-	2
Noreen [96]	2015	Pakistan	Women of childbearing age	430 (NS)	-	5	-	-	-
Osei [20]	2019	Ghana	Students	226 (100%)	-	-	56%	14%	30%
Park [19]	2012	South Korea	Women 30+ years old	4350 (NS)	-	-	-	40%	4
Park [18]	2013	South Korea	Men 40+ years old	2174 (NS)	-	-	-	33%	4
Rajamoorthy [10]	2019	Malaysia	General population >15 years old	764 (99%)	37%; 14.9 ± 3.8 ^a^	5	-	-	6
Roushan [97]	2013	Iran	General population >15 years old	13965 (87%)	-	6	-	-	-
Shakeel [98]	2015	Pakistan	General population >15 years old	434 (79%)	-	1	86%	33%	1
Vo [99]	2018	Viet Nam	Students	535 (NS)	3.5 ± 0.2 ^a^	6	-	-	-
Yang [100]	2015	China	Migrant worker	2065 (99%)	-	2	-	-	-
Zafrin [101] ††	2018	Bangladesh	General population >15 years old	-	Detailed per-question	6	-	-	-

† = Prospective cohort study; †† = Case-control study; HCWs = Health-care workers; ^a^ Mean (standard deviation); ^b^ Median (IQR).

**Table 3 vaccines-09-00625-t003:** The determinants of hepatitis B knowledge.

No	Factors	High-Risk Population	Low-Risk Population	Number of Studies *
**Sociodemographic factors**				
1	Age	Younger age group (positive association; ref = older age group) [11,12,13]; no association [15,16,17]	Younger age group (negative association; ref = older age group) [10]; no association [15,16,17]	4/7
2	Gender	Male sex (positive association; ref = women) [11,16]; no association [10,12,15]	No association [10,12,15]	2/5
3	Ethnic group	-	Malay ethnic group (positive association; ref = Indian ethnic group) [10]	1/1
4	Residency	Urban (positive association; ref = rural) [49,99]; no association [12,15]	No association [12,15].	2/4
5	Occupational status	Health-care worker (positive association; ref = unemployed) [17]	No association [10,15]	1/3
6	Monthly income	Higher income (positive association; ref = lower income) [10,13,15]	Higher income (positive association; ref = lower income) [10,13,15]	3/3
7	Level of education	Higher education (positive association; ref = lower education) [10,16,17,49]	Higher education (positive association; ref = lower education) [10,16,17,49]	4/4
**Work-related factors**				
8	Profession of HCW	Physician (positive association; ref = nurse/midwife/pharmacist) [11,49]; general practitioner (positive association; ref = specialist) [12]	-	3/3
9	Part-time job	No part time job (positive association; ref =having part-time job) [99]	-	1/1
**Student-related factors**				
10	Year of study	Higher level (positive association; ref = lower level) [99]	-	1/1
11	University/faculty/type of facility	Private facility (positive association; ref = public facility) [99]; no association [17,92]	-	1/3
**Experience factors**				
12	Knowing someone who lives infected	-	Yes (positive association; ref = no) [92]	1/1
13	Screening for Hepatitis B	Yes (positive association; ref = never) [11,49,92]; frequently/systematic (positive association; ref = never) [12]	Yes (positive association; ref = never) [11,49,92]	4/4
**Information exposure factors**				
14	Heard about hepatitis B/lecturer on hepatitis B	Yes (positive association; ref = never) [12,92]	Yes (positive association; ref = never) [12,92]	2/2
**Vaccination status**				
15	Vaccination status	Yes (positive association; ref = No) [99]; appropriate (positive association; ref = inappropriate) [11]	No association [49,92]	2/4

* Number of studies finding a significant association/number of studies investigating the fact.

**Table 4 vaccines-09-00625-t004:** The determinants of hepatitis B vaccine uptake.

No	Factors	High-Risk Population	Low-Risk Population	Number of Studies *
**Sociodemographic factors**				
1	Age	Older age group (positive association; ref = older age group) [15,29,77]; (negative association; ref = older age group) [13,19,52,90]; no association [14,17,18,20,67,70,75,102]	Older age group (positive association; ref = older age group) [15,29,77]; (negative association; ref = older age group) [13,19,52,90]; no association [14,17,18,20,67,70,75,102]	7/15
2	Gender	Female (positive association; ref = women) [20,28,66,80,102]; (Negative association; ref = women) [11]; No association [14,15,29,35,67,70,75,77,90]	Female (positive association; ref = women) [20,28,66,80,102]; no association [14,15,29,35,67,70,75,77,90]	6/15
3	Ethnic group	Lao Soung ethnic group (positive association; ref = Lao Loum ethnic group) [6]	-	1/1
4	Residency	Giansu (positive association; ref = Fijian) [90]; No association [14,15,19,20,66]	Urban (positive association; ref = metropolitan) [19]; no association [14,15,19,20,66]	2/7
5	Marital status/family status	Married (positive association; ref = single) [6]; with partner (positive association; ref = without partner [14]; single (positive association; ref = married [77]; no association [11,15,17,18,19,20]	No association [11,15,17,18,19,20]	3/8
6	Monthly income	No association [57]	Higher income (positive association; ref = lower income [15,18,19]	3/4
7	Health insurance	-	Having health insurance (positive association; ref = no [18,19]	2/2
8	Level of education/educational in year	Higher education (positive association; ref = lower education) [14,17,18,19,52]; no association [15,57,77]	Higher education (positive association; ref = lower education) [14,17,18,19,52]; no association [15,57,77]	5/8
9	Occupational status	Health worker (positive association; ref = unemployed) [17]	Routine and manual (positive association; ref = professional) [19]; teacher (positive association; ref = housewife) [15]; no association [18]	3/4
**Work-related factors**				
10	Profession of HCW	Laboratory staff (positive association; ref = nurse) [29]; internship doctor (positive association; ref = nurse/pharmacist/laboratory staff) [77]; medical technology/nurse (positive association; ref = physician) [90]; physician (positive association; ref = technician) [14]; nurse (positive association; ref = physician) [67]; nurse/consultant/resident (positive association; ref = house office) [75]; no association [11,35,102]	-	6/9
11	Work department	Outpatient department (positive association; ref = medical pediatric) [70]; high-risk department (positive association; ref = low-risk department) [90]; no association [35]	-	2/3
12	Work experience	10 years or less (positive association; ref = more than 10 years) [70]; less than 5 years (positive association; ref = 5 years and more) [35]; more than 10 years (positive association; ref = less than one year) [77]; no association [14,29,57,66,75,90,102]	-	3/10
13	Work regimen and level of satisfaction with the profession	Fixed (positive association; ref = hired employee) [52]; high satisfaction (positive association; ref = low satisfaction) [57]	-	2/2
14	Facility level	High level (positive association; ref = low level) [29]; tertiary hospital (positive association; ref = non-tertiary [17]; country/township hospital (positive association; ref = municipal) [90]; no association [29,67]	-	3/5
15	Management’s protection at workplace	Using personal protective equipment (positive association; ref = no) [14,80]; free hepatitis B vaccination from work (Positive association; ref = no) [90]; regular training in occupational health in the last two years (positive association; ref = no) [52]	-	4/4
**Student-related factors**				
16	Faculty	Post-graduation (positive association; ref = medicine); medicine (positive association; ref = basic science/pharmacy/medical technology) [6]	-	1/1
**Information exposure factors**				
17	Training infection	Yes (positive association; ref = No) [29,35,67,90,103]; no association [66]	-	5/6
**Experience factors**				
18	Exposure experience	Ever had experience of occupational exposure (positive association; ref = No) [35,52,57]; no blood transfusion history (positive association; ref = no) [80]; having positive family/friend of hepatitis B infected (positive association; ref = no) [28] no association [14]	-	5/6
19	Previous hepatitis B screening/anti-hepatitis B	Ever HBsAg screen test (positive association; ref = never) [11]; anti-hepatitis B status resulted positive (positive association; ref = resulted positive) [70]	-	2/2
**Knowledge**				
20	Hepatitis B knowledge	Acceptable knowledge (positive association; ref = unacceptable [90]; no association [20,29,66]	-	1/4
**Lifestyle**				
21	Alcohol consumption	Alcohol consumption (negative association; ref = no) [52,57,70]; no association [14,70]	-	3/5
22	Tobacco used	Tobacco used (negative association; ref = no) [57]; no association [14,70]	-	1/3

* Number of studies finding a significant association/number of studies investigating the topic.

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
