# Peer review of "Knowledge, Vaccination Status, and Reasons for Avoiding Vaccinations against Hepatitis B in Developing Countries: A Systematic Review"

_vaccines, 2021, doi:10.3390/vaccines9060625_

Round 1

Reviewer 1 Report

The systemic review done by Machmud et al. entitled "Knowledge, vaccination status, and reasons for avoiding vaccinations of hepatitis B in developing countries: A systematic review".

The review includes the following items: a) Electronic Database searches for HBV/vaccine/Health coworkers,..etc. Then the authors filtered the searches by removal of duplication, articles not meeting inclusion criteria , non English artciles, abstract conference,...etc 

b)  Risk of bias assessment:  Articles were divided into three categories of quality: unsatisfactory, satisfactory, and good study.  This is a good point.

c)Data analysis and synthesis: Independent variables such as age, gender, education, residency, marital status, monthly income, ethnicity, occupational status, and health insurance,...etc were considered.

d) Factor associated to knowledge and vaccination status

e) Reasons for not being immunized

f)Reasons for not being vaccinated

In general, the review is comprehensive and interrsting

I have few points to improve the qulaity of the manuscript

a) Please include in the discussion: vaccine escape mutation and how the HBV vaccination affect the emergence of spread opf the mutation

b) Correlate between the spread of vaccinee scape mutation with population criteria such as gender, age, occupational, and region...etc.

Author Response

Dear Reviewer 1,   

We thank the reviewer for the overall positive feedback on regarding our article “Knowledge, vaccination status, and reasons for avoiding vaccinations of hepatitis B in developing countries: A systematic review”. Please find the our response in each of their points.

Response to Reviewer 1 Comments

We thank the reviewer for the overall positive feedback on our manuscript.

Point 1: Reviewer 1 suggested adding some explanations for vaccine escape mutations and how hepatitis B vaccination affects the emergence of a mutation spread.

Response 1:

We have added this information in the discussion section (section 4.3: Reason for not being vaccinated, lines 94-101, page 25).

Point 2: Reviewer 1 also suggested reporting the associations between the spread of the vaccine escape mutation and population criteria such as age, gender, occupation, religion, etc.

Response 2:

As presented in the background section of our manuscript, the purpose of our study was to summarize the available evidence in order to identify predictors of the level of knowledge and vaccination status of hepatitis B, and reasons why people choose not to be vaccinated against hepatitis B in developing countries. As a result, the spread of the vaccine escape mutation was not in the focus of our manuscript. Apart from that, there is not enough scientific evidence about the association between the spread of the vaccine escape mutation and population criteria. Nevertheless, we added this issue as a study limitation in the discussion section (section 4.4: Strength and limitation of the review, lines 132-134, page 26). 

Sincerely,

Putri Bungsu Machmud & Prof Rafael Mikolajczyk (On behalf of the co-authors)

Reviewer 2 Report

The subject of research contributes to literature.

1. Developing country is not defined. According to what criteria was the definition of a developing country used? For example, Saudi Arabia, Turkey, South Korea has been evaluated in the category of developing country.

2. Suggestions on how the result of the research can be used by health policy makers will be useful.

3. A final grammar check will be useful.

Author Response

Dear Reviewer 2,   

We thank the reviewer for the overall positive feedback on regarding our article “Knowledge, vaccination status, and reasons for avoiding vaccinations of hepatitis B in developing countries: A systematic review”. Please find the our response in each of their points.

Response to Reviewer 2 Comments

We thank the reviewer for the overall positive feedback on our manuscript.

Point 1: Reviewer 2 raised concern about the used definition of a developing country, which included countries like Saudi Arabia, Turkey, and South Korea as developing countries.

Response 1:

Our definition of developing countries is based on the official report of the world economic situation and prospects 2020 from the United Nations. This report recognized that Saudi Arabia, Turkey, and South Korea are included in the category of developing countries. We have added this information about the origin of the used definition in our methods section (Section 2.2: Eligibility criteria section, lines 75-76, page 2).

Point 2: Reviewer 2 suggested to add explanations about how the study results can be used by health policy makers.

Response 2:

We have added this information in the conclusion section (lines 151-152, page 26).   

Point 3: Reviewer 2 also suggested to check the final grammar of this manuscript.

Response 3:

Our final manuscript was checked by native-speaking colleagues.

Sincerely,

Putri Bungsu Machmud & Prof Rafael Mikolajczyk (On behalf of the co-authors)

Round 2

Reviewer 1 Report

I checked the revised manuscript submitted by the authors and the review could be published in the current form